# You Are What You Eat and So Is Our Planet: Identifying Dietary Groups Based on Personality and Environmentalism

**DOI:** 10.3390/ijerph19159354

**Published:** 2022-07-30

**Authors:** Jan-Felix Palnau, Matthias Ziegler, Lena Lämmle

**Affiliations:** 1Department of Psychology, Faculty of Life Sciences, Humboldt-Universität zu Berlin, 10117 Berlin, Germany; zieglema@hu-berlin.de; 2Department of Psychology, Faculty of Human Sciences, Medical School Hamburg, 20457 Hamburg, Germany; lena.laemmle@medicalschool-hamburg.de

**Keywords:** behavioral change intervention, environmentalism, dark triad, Big Five, plant-based diet, meat consumption, meat attachment, food neophobia, consumption orientations, segmentation analysis

## Abstract

Behavioral change interventions promoting the reduction of animal product consumption are valuable tools to improve ecological sustainability as well as public health and help the mitigation of climate change. Recent findings revealed improved efficacy of interventions targeted at barriers (e.g., self-efficacy) of three different types of meat consumers over non-targeted interventions (e.g., completion of unrelated surveys). However, such interventions have yet to factor in the role of individual differences in personality. Therefore, in a first step, we performed segmentation analysis on barriers and benefits of reducing animal product consumption (e.g., meat attachment, environmentalism) with the inclusion of personality. In an online sample of N=1135 participants, latent profile analysis revealed five distinct dietary groups: “plant-based eaters”, “meat-reducers”, “medium-hindrance meat eaters”, “medium strong-hindrance meat eaters, and “strong-hindrance meat eaters”, based on inhibitors and facilitators of meat reduction. Groups differed in terms of consumption of different animal products ($\eta^{2}=0.08$ to $\eta^{2}=0.80$) as well as the Big Five ($\eta^{2}=0.08$ to $\eta^{2}=0.80$) and Dark Triad ($\eta^{2}=0.08$ to $\eta^{2}=0.80$). Strong-hindrance meat eaters were characterized by low Conscientiousness, Agreeableness, and Openness as well as high dark trait expression, implying new targets for future intervention design.

## 1. Introduction

While subsidized policies in line with emission targets set by the European Commission [1] play a crucial role in climate change mitigation, individual behavioral change towards a low-carbon society also represents an important step to achieve the overall goal [2,3]. Besides recycling [4], the adoption of a plant-based diet is highlighted as the most effective consumer-side intervention [5,6,7,8]. First results of behavior change interventions promoting the reduction of animal product consumption showed improved efficacy of interventions (+40 g daily CO_2_e reduction) targeted at different types of meat eaters by implementing group-specific behavioral change strategies (e.g., social norms and suggestion of red meat substitution for strong-hindrance meat eaters) over non-targeted ones [9]. However, such interventions have yet to target stable individual differences. Thus, we initially aimed to expand recent results of segmentation research on dietary choices [10] by the inclusion of personality. Explicitly, latent profile analysis (LPA) will be performed based on the most relevant barriers to meat reduction used in previous studies (environmentalism, meat attachment, self-efficacy, food neophobia, social conformity, health orientation) in order to identify segments of consumers. We choose to then investigate differences in personality, because of existing links between the Big Five and Dark Triad [11] and meat consumption [12,13]. The Big Five Model comprises five personality traits: Openness to experience, Conscientiousness, Extraversion, Agreeableness, Neuroticism. While people open to experience are imaginative and curious, Conscientiousness is characterized by a tendency to be disciplined and organized. Furthermore, Extraversion is described by high sociability and assertiveness, whereas agreeable people are trusting, helpful and get along well with others. Finally, Neuroticism is marked by a tendency to experience negative emotions (e.g., distress, anxiety) and cognitively exhibit pessimism e.g., [14]. In contrast, the Dark Triad specifies three dark personality traits, i.e., narcissism, Machiavellianism, and psychopathy, characterized by social aversiveness [15]. Narcissists seek ego-reinforcement and hold a grandiose sense of self-entitlement. Moreover, Machiavellianism is characterized by strategic manipulativeness in order to achieve one’s instrumental goals. While psychopaths share the goals of people high in Machiavellianism, they are distinctively impulsive and affectively callous [16]. Aside from personality, food consumption orientations, conceptualized as people’s main motivations to consume foods [17], were added to the set of indicators in order to investigate how personality relates to different motives of food choices, regarding both animal products and plant-based foods. Dietary groups identified based on these constructs could then be targeted by (online) interventions by matching persuasive appeals to personal needs [18]. To counter the environmental consequences of climate change, which currently constitutes the biggest global threat, significant reduction of carbon dioxide (CO_2_) emissions is required [19]. Beyond meat consumption’s contributions to global warming [20], reduction of all animal products (meat, fish, eggs and dairy) can help to lower an individual’s total carbon footprint by up to 22%, outperforming other household actions [21]. Yet, an overemphasis of individual behavior in the mitigation process of climate change may distract from the need for systemic changes, particularly in the energy sector as the biggest source of CO_2_ emissions [22,23,24]. However, beyond carbon emissions, plant-based diets could have transformative potential related to the sustainability challenge of global food supply by reducing food’s land use by up to 76% [25]. In addition, higher consumption of plant-based foods is also associated with lower risk of disease and lower mortality [26,27]. Thus, the conception of behavioral change interventions promoting plant-based eating habits are potentially highly valuable in both improving sustainability and public health as well as helping the mitigation of climate change. While the effectiveness of behavioral change interventions targeting the reduction of meat consumption is well documented [4,28], the reduction of other animal products (e.g., eggs and dairy) has been investigated only recently. Initially, Lacroix and Gifford [10] identified three different types of meat-consumers (reducer, moderate-hindrance, strong-hindrance) based on an individual’s capability, opportunity, motivation (COM-B) [29], habit strength [30], change state [31], and perceived barriers and benefits [32] using LPA. However, personality has yet to be considered, despite it being highlighted as a main driver of change in health behavior [33]. Explicitly, people’s responsiveness to intervention techniques differs between personality traits in the context of gamified intervention systems [34]. Whereas higher trait expression of all the Big Five, expect Neuroticism, is linked to higher persuasiveness by personalization of a health intervention, mainly more extraverted and conscientious people are motivated by goal setting. Moreover, the motivation of extraverted and agreeable people is increased by comparison, punishment, competition and rewards, while those intervention techniques decrease motivation in more open individuals.

### 1.1. Segmentation Studies of Dietary Choices

Segmentation techniques have only sparsely been used to study meat consumption, although an increasing body of research has grown in the last years. As part of a choice experiment, Apostolidis and McLeay [35] determined six types of consumers with different socio-demographic characteristics and meat consumption habits utilizing latent class analysis: price conscious, healthy eaters, taste driven, green, organic and vegetarian. Likewise, six groups have been identified through segmentation based on consciousness of meat consumption’s environmental effects using cluster analysis [36]. Revealed types ranged from “highly conscious” to “resistant”, with “highly unsure” consumers being the most prevalent. As expected, members of the conscious clusters showed higher willingness to reduce meat consumption, opposed to those of the resistant cluster. Furthermore, latent class analysis found six groups in data of self-reported beef, bean and soy consumption [37]. Contrasts revealed higher concerns regarding nature, weight and health of the “no beef” group contrary to the “only beef” group. A more exploratory approach examined the representations and impact of meat, as well as reasoning for the change of meat consumption [38]. Based on this, three clusters of consumers (“disgust towards meat and moral internalization”, “low affective attachment and willingness to change”, and “attachment and unwillingness to change”) have been identified. We expect to extract a similar quantity of dietary groups from our data, although inclusion of personality could also result in a more complex segment structure. However, identified segments were merely used to evaluate the efficacy of tailored interventions [9]. Remaining evidence points to improved efficacy of interventions with instructional information on how to reduce meat consumption tailored to the change state [39]. With the consideration of meta-analytical evidence on tailoring [40], we also expect higher efficacy of interventions promoting plant-based eating habits targeted at barriers and benefits, including personality.

### 1.2. The Nomological Network of Eating Behavior

#### 1.2.1. Big Five

Research on food choice and personality is still in its early stages. Notably, people high in Openness were consistently found to eat less meat and more plant-foods (e.g., [41]; see Table 1), whereas recent results indicate links to all other traits as well [42,43]. With food choices categorized as health behavior (e.g., [44]), higher Conscientiousness, higher Extraversion and lower Neuroticism also correspond to better mental, physical, and sexual health states altogether (e.g., [45,46,47]). 

#### 1.2.2. Dark Triad

Likewise, while the relation between eating behavior and the Dark Triad has not explicitly been investigated yet, dark traits are also linked to health outcomes [48] and general health-related behaviors [49], with psychopathy being the strongest predictor of negative manifestations in both instances and narcissism correlating with positive health outcomes overall [50]. With regard to environmentalism, recent findings indicate a link between communal narcissism and pro-environmental behavior, while high agentic trait expression corresponds to low altruistic narcissism [51].

**Table 1 ijerph-19-09354-t001:** Links between the Big Five and eating behavior.

Authors	Findings
Pfeiler & Egloff, 2018 [52]	links between following a vegetarian or vegan diet and higher Openness
Pfeiler & Egloff, 2018 [53]; Pfeiler & Egloff, 2018 [41]	increased meat consumption of people low in Openness, Conscientiousness and agreeableness
Pfeiler & Egloff, 2018 [41]	lower Openness corresponding to more red meat, higher Openness to more fish consumption
Pfeiler & Egloff, 2020 [43]	consumption of carbohydrate-based foods associated with high Conscientiousness, agreeableness and emotional stability consumption of meat-based foods associated with lower trait expression of all Big Five except Extraversion consumption of plant-based foods associated with higher trait expression of all Big Five
Hopwood & Bleidorn, 2019 [42]	people labeling meat as “necessary” and “normal”: higher in Neuroticism, but lower in Openness and agreeableness

#### 1.2.3. Environmentalism

Although a great deal of research has been conducted on environmentalism itself and meat consumption’s detrimental effects on climate change are well documented (see [8,20,54]), only few studies have investigated the link between environmental attitudes and food choices. Existing evidence shows that individuals with pro-environmental attitudes eat less meat [53] and that the ones who limit meat intake for ecological reasons, tend to be female, young, and meat-reducers rather than vegetarians or vegans [55]. With respect to personality, high Openness corresponds to both pro-environmental attitudes and behavior on trait level as well as most facet levels [56]. The authors’ findings also indicate similar associations with agreeableness and Conscientiousness. Recent findings not only indicate that pro-environmental attitudes translate into pro-environmental behavior (PEB), but also mediate the relationship between the trait honesty-humility and PEB.

#### 1.2.4. Meat Attachment

High meat attachment, as positive representations of meat [57] corresponds to following a meat-based or omnivorous diet, whereas low meat attachment is linked to vegetarian or vegan eating habits [57]. Beyond that, high correlations with positive attitudes towards meat have been found. Moreover, people high in meat attachment hold more subjective norms and attributes of human supremacy but exhibit less environmental concerns and willingness to reduce meat consumption [58].

#### 1.2.5. Food Neophobia

Food neophobia is characterized by an aversion towards new foods [59] and is highlighted as a key barrier in acceptance of plant-based protein sources [60]. With regard to personality, food neophobia has been linked to lower Openness to experience [61,62,63]. Additionally, neophilics (approach factor) tend to be more open and agreeable, whereas those who avoid new foods (avoidance factor) are generally lower in Openness and Extraversion [64]. Going beyond the negative association between food neophobia and liking of foods overall [65], people with higher food neophobia consume smaller amounts of vegetables, salad and poultry, while also reporting a lower liking for salty snacks and whole grain bread. Moreover, meat neophobia was found to generally be higher than plant neophobia [62]. Findings also suggest gender differences, with women showing higher meat neophobia.

#### 1.2.6. Food Consumption Orientations

General consumption orientations refer to shared understandings of products and services regarding their acquisition, appropriation, appreciation and disposal [66], whereas food consumption orientations specifically represent the primary motivations and justifications held by people concerning their food consumption [17]. As to be expected, food consumption orientations hold greater predictive ability of plant and meat consumption [67], with health- and naturalness-concerning orientations being the strongest predictors. Food consumption orientations about health and naturalness associate with lower meat consumption and a more plant-based diet, respectively. Additionally, they also predict willingness to follow a plant-based diet and reduce meat consumption. On the other hand, an orientation towards pleasure hinders the change of eating habits.

### 1.3. Aims of the Study

The first goal of this study was to identify dietary groups based on inhibitors of meat reduction and food consumption orientations using LPA in a sample adequately sized for its computation (see Section 2.1, [68]) in order to investigate whether commonly found solutions (e.g., three profile solution from Lacroix and Gifford, [10]) can be reproduced. Secondly, our main research aim was to describe the personality profiles of those groups. With the description of those groups regarding inhibitors, food consumption orientations and personality we aimed to provide further leverage points for future interventions targeting the consumption of animal products.

## 2. Materials and Methods

### 2.1. Participants and Procedure

Data for this study were collected by an online questionnaire as part of a nation-wide survey. Participants aged 18 and above were recruited via online recruitment systems of the Humboldt-Universität zu Berlin and Medical School Hamburg (MSH) and social media (sharing via messaging services, e.g., WhatsApp, Signal). Although distance between classes and accurate model selection have shown greater impact on the outcome of statistical power than sample size, the maximum size tested in simulation studies on LPA (*N* = 1000) was set as a minimum to acquire [68]. Participants were briefed about the anonymity and voluntariness of their participation and informed consent was obtained. Upon survey completion, participants were thanked, and students were granted course credit for their participation. Ultimately, 1300 participants completed the survey. After removal of observations with |z|>3 on any scale entering analyses [69], a final sample of N=1135 remained. The sample’s mean age was 27 years (SD=8.91). *N =* 855 (75.33%) participants identified as female, 271 (24.66%) as male, 9 (0.01%) as diverse. Regarding dietary identity, 726 (63.94%) participants reported being omnivorous, 324 (28.55%) vegetarian, and 85 (7.51%) vegan.

### 2.2. Measures

#### 2.2.1. Demographic Variables

Gender, age, and dietary self-identity (omnivorous, [ovo-lacto/pesco]-vegetarian, vegan) were measured and included as outcome variables.

#### 2.2.2. Eating Behavior

Based on existing links between the Big Five and consumption of different food types, we choose to include consumption of all food categories (e.g., vegetables, fruits, meat, dairy products, bread, sweets) instead of just animal products [9]. Regarding prepared foods, we also differentiated between plant-based and non-plant-based food items (plant-based vs. non-plant-based sweets). With this detailed approach, we expect to find more differentiated and hence more targetable profiles (e.g., high-hindrance meat eaters with low vs. high vegetable consumption) than previous studies have shown. We used a 50-item measure developed for the purpose of this study based on inputs from Haftenberger et al. [70] and Pfeiler and Egloff [43] to assess eating behavior of the last 30 days. The measure used an 8-point scale (1 = “never” to 8 = “two or more times a day”) to help prevent ceiling or floor effects and comprises the food types: milk (“How often did you drink milk?”), alcoholic beverages, non-alcoholic beverages, vegetables, fruits, bread, butter and margarine, cheese, quark or yogurt, sweet spreads, savory spreads, eggs, potatoes, legumes or pulses, red meat, processed meat products, poultry, fish, meat replacements, cake or sweet pastries, confectionery, breakfast cereals, snack foods. Additionally, each food category included an item asking for the share of plant-based food items (e.g., “How often did the sweets contain animal products (e.g., eggs, butter, milk)?”) with a 5-point scale (1 = “never” to 5 = “always”).

#### 2.2.3. Big Five

The Big Five were measured with the 30-item German short version of the BFI-2 [71] with a 5-point rating scale (1 = “totally disagree” to 5 = “totally agree”). The NEO-FFI-30 includes the dimensions of Neuroticism (e.g., “I get stressed out easily”, 6 items), Extraversion (e.g., “I am the life of the party”, 6 items), Openness (e.g., “I have a vivid imagination”, 6 items), Conscientiousness (e.g., “I get chores done right away”, 6 items) and agreeableness (e.g., “I treat everyone with kindness and sympathy”, 6 items). The psychometric properties of the BFI-2-S are similar to the 60-item version with internal consistencies ranging from α=0.65 to α=0.80 (i.e., α=0.69 to α=0.81 in our sample).

#### 2.2.4. Dark Triad

To assess the Dark Triad, we used the 27-item Short Dark Triad in its German adaption [11] with the same 5-point rating scale. The Short Dark Triad targets Machiavellianism (e.g., “It’s not wise to tell your secrets”, 9 items), narcissism (e.g., “People see me as a natural leader”, 9 items) and psychopathy (e.g., “I like to get revenge on authorities”, 9 items). The measure exhibits acceptable reliability estimates with values between α=0.72 and α=0.85 (i.e., α=0.70 to α=0.76 in our sample).

#### 2.2.5. Environmentalism

Environmentalism was measured with the German form of the New Ecological Paradigm (NEP) Scale [72,73]. The measure consists of 15 items assessing the degree of pro-environmental orientation on a 5-point rating scale (1= “strongly disagree” to 5 = “strongly agree”). The internal consistency (α=0.83) is satisfactory ([72]; α=79 in our sample).

#### 2.2.6. Meat Attachment

For the assessment of meat attachment, we used a German translation of the Meat Attachment Questionnaire [38], consisting of 16 items with a 5-point rating scale ranging from 1 = “strongly disagree” to 5 = “strongly agree”. Complementing Lacroix and Gifford [9,10] we utilized all four subscales of hedonism (e.g., “To eat meat is one of the good pleasures in life”, 4 items), affinity (e.g., “By eating meat I’m reminded of the death and suffering of animals”, 4 items), entitlement (e.g., “According to our position in the food chain, we have the right to eat meat”, 4 items) and dependence (e.g., “Meat is irreplaceable in my diet”, 4 items), which can be subsumed to the second-order global dimension of meat attachment. Subscales’ internal consistencies range from α=0.77 to α=0.90 with a score of α=0.92 for the global scale [38]. Its’ German adaption was used [74], which showed good reliability in our sample (α=0.87).

#### 2.2.7. Food Neophobia

To measure food neophobia we used the 10-item German version of the Food Neophobia Scale (e.g., “I do not trust new foods”; [65,75]; Siegrist et al. 2013) with a 7-point rating scale (1 = “do not agree at all” to 7 = “fully agree”). The overall internal consistency was good (α=0.83).

#### 2.2.8. Food Consumption Orientations

To cover additional barriers and benefits measured by Lacroix and Gifford (e.g., conformity, healthy-eater identity, food involvement, [9,10]), tailored all food categories instead of just meat consumption, we assessed food consumption orientations using the German Brief Eating Motivation Survey [17]. The measure consists of 45 items in total with a 5-point scale (1 = “totally disagree” to 5 = “totally agree”) and includes the dimensions of liking (e.g., “because it tastes good”, 3 items), habits (e.g., “because I usually eat it”, 3 items), need & hunger (e.g., “because I need energy”), health (e.g., “Because it is healthy”, 3 items), convenience (e.g., “Because it is quick to prepare”, 3 items), pleasure (e.g., “In order to indulge myself”, 3 items), traditional eating (e.g., “because I grew up with it”, 3 items), natural concerns (e.g., “Because it is organic”, 3 items), sociability (e.g., “so I can spend time with other people”), price (e.g., “because it is inexpensive”, 3 items), visual appeal (e.g., “because the presentation is appealing”, 3 items), weight control (“ because it is low in calories”, 3 items), affect regulation (e.g., “Because I am sad”, 3 items), social norms (“because it would be impolite not to eat it”, 3 items), and social image (e.g., “Because it makes me look good in front of others”, 3 items). Internal consistencies of the subscales were acceptable, ranging from α=0.67 to α=0.90. The dimension of need and hunger was not included in analyses due to its low reliability (α=0.61). With the assessment of consumption orientations towards price, we followed the proposal to consider both habit strength and affordability as a contextual factor of reducing animal product consumption and [10,57].

#### 2.2.9. Self-Efficacy

Self-efficacy as related to the reduction of animal product consumption was assessed using a four-item scale (e.g., “I lack the cooking skills to prepare meat-free meals”), created by Lacroix & Gifford [10], with a 5-point scale (1 = “strongly disagree” to 5 = “strongly agree”). The measure’s reliability is satisfactory (α=0.70). Items of the original measure were translated into German, translated back into English and retranslated into German.

#### 2.2.10. Willingness to Change Eating Habits

Analogous to Graça et al. [38,57], participants were asked to report their willingness to reduce meat consumption, to reduce animal product consumption in general and to follow a purely plant-based diet with a single item each (“Please indicate your willingness to: (1) reduce meat consumption, (2) reduce other animal product consumption, (3) follow a plant-based diet”), using a scale in the range of 1 = “not willing at all” and 5 = “very willing”). With that, we expect to gain insight into segments’ willingness for dietary change in the context of personality.

### 2.3. Statistical Analyses

Data was analyzed using R in RStudio [76,77]. In order to determine different types of consumers in our data, we performed latent profile analysis (LPA) using mclust package (version 5.4.9, L., Scrucca, sourced from CRAN repository in RStudio [78]), for which all constructs were included as segmentation variables in the base model, with the exception of demographic variables, food orientation towards hunger, and food neophobia. Because of missing values due to technical errors, food neophobia was added to outcome variables instead. Aside from meat attachment’s global scale, every subscale entered analysis. We chose to perform LPA, since part of our exploratory approach is the inclusion of a wide range of, partially overlapping, variables (e.g., Dark Triad and agreeableness or Openness and food neophobia), which made multicollinearity a valid concern to be dealt with. Cluster analysis may skew results in favor of those variables highly correlated, whereas multicollinearity is less of a concern in LPA due to its model-based approach and reliance on probabilities in order to identify the optimal numbers of segments to extract [79]. In addition, LPA provides a more consistent procedure, which prevents subjective methodology and allows for higher replicability [80]. Regarding LPA, we then compared models to (1) identify the best fitting model and (2) extract the optimal number of profiles. Line-plotting of the extracted profiles allowed for an intelligible visualization. To assess model fit, Bayesian information criterion (BIC), entropy and bootstrapped likelihood ratio test (BLRT) were considered [79]. Akaike’s information criterion (AIC) was excepted from the evaluation of model fit, since it performs poorly at selecting the correct number of profiles regardless of sample size, degree of separation between those, or quantity of indicators [68]. Beyond this, part of a heuristic reasoning process was also the sighting of external material, in this case socio-demographic characteristics, as they were not included as segmentation variables.

Since assumptions of multivariate analysis of variance (MANOVA) were not met, we then performed one-way analyses of variance (ANOVA) using Welch’s F and post-hoc tests controlling for false discovery rate (FDR; Benjamini–Hochberg correction) in order to examine mean differences between identified segments in both profiling and outcome variables as well as personality traits [81]. All data, scripts, and the preregistration manuscript of our first methodological draft are publicly available in an Open Science Framework (OSF) repository (https://osf.io/xewh8/?view_only=e1e32f2dd63841e1bd07d2f27d9d9dd9 [accessed on 13 July 2022]).

## 3. Results

### 3.1. Latent Profile Analysis

Initially, LPA was performed with inclusion of all indicator variables (all inhibitors and food consumption orientations). However, on that initial set, *mclust* was unable to calculate less parsimonious models (e.g., VVE, VVV) for more than two profiles due to the number of parameters needed to be estimated [78]. Since we expected assumptions of local independence and homogeneity of variance-covariance matrices to be violated across classes, we opted against the retainment of only parsimonious models (e.g., EEE, EEI), which assume equal variance as well as equal and no covariance between profiles, respectively. Specifically, the inclusion of food orientation towards social image prevented computation of more complex models for a higher number of profiles. Therefore, we removed it from the set of indicators, which yielded an improvement in model estimation.

We then assessed model fit based on BIC and evaluated the interpretability of profile solutions based on differences in outcomes (e.g., consumption of animal products). The model fit indices are presented in Table 2.

Ultimately, highest BIC values were obtained for the five profile model with VVE parameterization. In *mclust* a higher BIC indicates better fit [78]. Distinctiveness of profiles was sufficient based on entropy values for all profile solutions with VVE parameterization, but BIC slightly favored the five-profile solution over the three- and six-profile models. Although BLRT suggested an improvement of the six-profile solution, we retained the five-profile model, since the BIC performs best at selecting the correct number of profiles [82]. While the fifth profile only had a small number of participants (5% of the sample), it provided great value in terms of interpretability and theoretical reasoning.

Regarding profile membership, 221 participants (19.5%) were assigned to the “plant-based eaters”, 211 participants (18.5%) to the “meat-reducers”, 416 participants (36.6%) to the “medium-hindrance meat eaters”, 232 participants (20.4%) to “medium-strong-hindrance meat eaters”, and 55 participants (5%) to the “strong-hindrance meat eaters”.

One-way ANOVAs using Welch’s F revealed significant mean differences between groups for all profiling and outcome variables except consumption of other dairy products (i.e., cheese), with effect sizes ranging from small to large ($\eta^{2}=0.08$ to $\eta^{2}=0.86)$.

### 3.2. Inhibitors and Food Consumption Orientations

One-way ANOVAs using Welch’s F revealed significant mean differences between groups for all profiling variables, with effect sizes ranging from medium to large ($\eta^{2}=0.13$ to $\eta^{2}=0.86)$. Both means of profiling variables and results of ANOVAs as well as post-hoc tests are presented in Appendix A Table A1 and Table A2, respectively. Meat attachment was lowest for the plant-based eaters and highest for the strong-hindrance meat eaters. In comparison, reported pro-environmental attitudes as well as self-efficacy (i.e., cooking skills to prepare plant-based meals) decreased from the highest for plant-based eaters to the lowest for strong-hindrance meat eaters.

The plant-based eaters showed highest orientation towards healthiness, but not naturalness of foods and were less motivated by traditional eating. In contrast, plant-based eaters were highly oriented towards pleasure (i.e., indulging in food) and liking of foods, but reported low affect regulation as a motive for food choices. Additionally, orientation towards convenience (i.e., low difficulty of food preparation) was high for them. Expression of motivation by sociability (i.e., eating to spend time with others) was similarly high to the other groups, with exception of the medium-strong-hindrance meat eaters. Reported orientation towards price (i.e., inexpensiveness of food) and weight control (i.e., preference for low calorie foods) was low compared to the strong-hindrance groups, while selection of foods used for weight loss was low compared to those. Compared to the plant-based eaters, meat-reducers were similarly high in environmentalism, self-efficacy, orientation towards naturalness, health, tradition and weight, but reported higher meat attachment and orientation towards pleasure of foods. They also scored lowest on food choices based on appetite, affect, price and social norms out of all groups.

The strong-hindrance meat eaters showed higher meat attachment and lower pro-environmental attitudes as well as self-efficacy compared to the meat-reducers. Moreover, they were most concerned about pleasure, convenience, and social norms of food. Likewise, strong-hindrance meat eaters reported a high orientation towards affective regulation and tradition and the highest orientation towards sociability.

The medium-strong-hindrance meat eaters exhibited highest meat attachment, while the strong-hindrance meat eaters were least self-efficient and least pro-environmentally oriented. Both groups reported a lower orientation towards health and naturalness of foods compared to the other groups, with the strong-hindrance meat eaters scoring lowest for healthiness and medium-strong-hindrance meat eaters lowest for naturalness. Individuals in both groups reported their eating habits being highly motivated by tradition, with the strong-hindrance meat eaters being the highest. Regarding orientation towards affect regulation, its expression was high for medium-strong-hindrance meat eaters, but low for the strong-hindrance group, while the opposite was found for orientation towards liking of food. Finally, strong-hindrance meat eaters were also highest in orientation towards price, appetite, weight control, and social norm.

### 3.3. Outcomes

z-standardized means of the outcome variables are displayed in Figure 1. Following the results of post-hoc tests with Benjamini–Hochberg correction for food neophobia, mean differences between all groups except the pairs of plant-based eaters and meat-reducers as well as medium-hindrance meat eaters and medium-strong-hindrance meat were significant with small to large effect sizes (F(4, 158)=16.44, p<0.001, $\eta^{2}=0.29$; $M_{diff}=0.20$, p<0.05, d=0.36 to $M_{diff}=0.82$, p<0.001, d=1.44). Moreover, all groups aside from MSHM and strong-hindrance meat eaters significantly differed in their consumption of poultry, red and processed meat (F(4, 285)=199.33, p<0.001, $\eta^{2}=0.74$ to F(4, 282)=277.21, p<0.001, $\eta^{2}=0.80$; $M_{diff}=0.41$, p<0.001, d=0.33 to $M_{diff}=2.40$, p<0.001, d=2.45).

Additionally, mean differences in fish and egg consumption were significant with large effect sizes $(F\left(4,\;299\right)=68.31,\;p<0.001,\eta^{2}=0.48;F\left(4,\;301\right)=26.04,p<0.001,\;\eta^{2}=0.26$), but only the plant-based eaters differed significantly from all other groups ($M_{diff}=0.88$, p<0.001, d=0.63 to $M_{diff}=1.52$, p<0.001, d=1.29). Regarding milk consumption, mean differences between plant-based eaters and all groups except strong-hindrance meat eaters were significant ($F\left(4,\;306\right)=6.58,p<0.001,\eta^{2}=0.08;\;M_{diff}=0.80$, p<0.05, d=0.29 to $M_{diff}=1.13$, p<0.001, d=0.48). In addition, all five groups were significantly different in their consumption of meat replacements and plant-based replacements for other products (e.g., vegan cheese and sweets; $F\left(4,\;302\right)=38.32,p<0.001,\;\eta^{2}=0.34$, $\eta^{2}=0.59$; $M_{diff}=0.43$, p<0.01, d=0.28 to $M_{diff}=1.89$, p<0.001, d=1.87). The only exception were the strong-hindrance meat eaters, which did not differ significantly from the plant-based eaters, meat-reducers and medium-hindrance meat eaters regarding meat replacements and from the meat-reducers for other plant-based replacements, respectively. Furthermore, results on differences in the groups’ willingness to give up meat, other animal products, and adopt a plant-based diet showed large effects ($F\left(4,\;297\right)=57.61,p<0.001,\eta^{2}=0.44$ to $F\left(4,\;307\right)=99.21,p<0.001,\eta^{2}=0.56$; see Figure 2). Finally, dietary identy (omnivore, vegetarian, vegan) significantly differed between groups (F(4, 297)=138.96,p<0.001, $\eta^{2}=0.65$), with large effect sizes for differences between plant-based eaters and the other groups and small to medium effects for other pairings ($M_{diff}=0.15$, p<0.001, d=0.37 to $M_{diff}=1.04$, p<0.001, d=2.21).

### 3.4. Personality

ANOVAs showed significant differences between groups for all of the Big Five (F(4, 309)=6.58, p<0.01, $\eta^{2}=$ 0.08 to F(4, 308)=22.45, p<0.001, $\eta^{2}=$ 0.23). Based on inputs from Gignac and Szodorai [83], we consider Cohen’s *d* of 0.2, 0.41, and 0.63 as small, medium, and large in the context of personality-specific effects. Specifically, for Extraversion the meat-reducers showed significantly higher trait expression than the medium- to strong-hindrance meat eaters ($F\left(4,\;301\right)=6.58,p<0.001,\eta^{2}=0.08;\;M_{diff}=0.26$, p<0.01, d=0.41 to $M_{diff}=0.28$, p<0.05, d=0.46; see Figure 3). Mean differences for agreeableness were significant between all groups ($F\left(4,\;305\right)=16.16,p<0.001,\eta^{2}=0.17;\;M_{diff}=0.15$, p<0.05, d=0.27 to $M_{diff}=0.52$, p<0.001, d=1.05), with the exception of the plant-based eaters not differing from the meat-reducers and strong-hindrance meat eaters, from which the medium-hindrance meat eaters did not differ either. Results revealed the same pairwise pattern for Conscientiousness $(F\left(4,\;308\right)=12.41,p<0.001,\eta^{2}=0.14;\;M_{diff}=0.19$, p<0.05, d=0.27 to $M_{diff}=0.54$, p<0.001, d=0.89). Furthermore, plant-based eaters were more neurotic than meat-reducers and strong-hindrance meat eaters ($F\left(4,\;315\right)=14.11,p<0.001,\eta^{2}=0.14;\;M_{diff}=0.35$, p<0.001, d=0.47; $M_{diff}=0.21$, p<0.05, d=0.28) and meat-reducers less neurotic than medium- and strong-hindrance meat eaters ($M_{diff}=0.44$, p<0.001, d=0.57; $M_{diff}=0.32$, p<0.01, d=0.49). Additionally, medium-hindrance meat eaters showed higher Neuroticism than the medium-strong-hindrance group ($M_{diff}=-0.29$, p<0.001, d=0.40). Plant-based eaters and meat-reducers were more open to experience than all other groups ($F\left(4,\;301\right)=26.04,p<0.001,\eta^{2}=0.23;\;M_{diff}=0.27$, p<0.001, d=0.40 to $M_{diff}=0.60$, p<0.001, d=1.02), but did not differ significantly from each other. Lastly regarding Openness, medium-hindrance meat eaters also differed significantly from the strong-hindrance meat eaters ($M_{diff}=0.28\;$, p<0.01, d=0.45).

Regarding the dark triad, ANOVAs revealed significant mean differences between profiles for all dark traits (F(4, 315)=7.68, p<0.001, $\eta^{2}=$ 0.09 to F(4, 308)=31.21, p<0.001, $\eta^{2}=$ 0.29). Plant-based eaters and meat-reducers both showed low expression of Machiavellianism compared to the medium- and medium-strong-hindrance meat eaters ($M_{diff}=0.29$, p<0.001, d=0.50 to $M_{diff}=0.35$, p<0.001, d=0.62), although both pairings did not differ significantly from each other. Finally, the strong-hindrace meat eaters were most Machiavellian with moderate to large effects of pairwise comparisons $(F\left(4,\;308\right)=31.21,\;p<0.001,\;\eta^{2}=0.29;$ $M_{diff}=0.31$, p<0.001, d=0.54 to $M_{diff}=-0.65$, p<0.001, d=1.28). With further respect to narcissism, the strong-hindrance meat eaters exhibited highest trait expression $(F\left(4,\;315\right)=7.68,\;p<0.001,\;\eta^{2}=0.09;$ $M_{diff}=0.29$, p<0.001, d=0.55 to $M_{diff}=0.35$, p<0.001, d=0.70), whereas all other groups were similar and the plant-based eaters being the lowest in narcissism, respectively. Concerning psychopathy, plant-based eaters and meat-reducers showed similarly low expression and both showed lower psychopathy than the medium-hindrance meat eaters $(F\left(4,\;300\right)=29.00,\;p<0.001,\;\eta^{2}=0.28;$ $M_{diff}=0.13$, p<0.05, d=0.26; $M_{diff}=0.20$, p<0.001, d=0.40). Meat-reducers were also significantly less psychotic than the medium-strong-hindrance meat eaters ($M_{diff}=0.16$, p<0.01, d=0.33). Ultimately, the strong-hindrance meat eaters were most psychopathic with mean differences reaching high effect sizes ($M_{diff}=0.64$, p<0.001, d=1.18 to $M_{diff}=0.84$, p<0.001, d=1.65).

## 4. Discussion

### 4.1. General Discussion

This study built on prior research on segmentation analysis that revealed three types of consumers based on meat consumption’s inhibitors: meat-reducers, medium- and strong-hindrance meat eaters [9]. We performed latent profile analysis in order to identify dietary groups based on those core inhibitors and food consumption orientations in a sample of N=1035 Germans. Results of mixture modeling suggested a five instead of a three-profile solution, providing greater insight into previously identified ones. In due of theorical considerations, we divided the sample into five groups: “plant-based eaters”, “meat-reducers”, “medium-hindrance meat eaters”, “medium-strong-hindrance meat eaters”, and “strong-hindrance meat eaters”. Group membership successfully explained a large amount of variance of profiling and outcome variables (i.e., consumption animal products and plant-based replacements).

Our five-profile solution adds to previous research examining groups of meat consumers with a comparable segmentation approach [10,57], in that it provides a more detailed differentiation between those groups previously identified. This further distinction would have been lost within a three-profile solution but reveals the plant-based eaters to be an additional low-hindrance group besides the meat-reducers (ours are similar to the meat-reducers from Lacroix & Gifford [10]). Accordingly, the plant-based eaters, showed lowest expression on all inhibitors and outcomes. The plant-based eaters, primarily vegetarians and vegans, seemed to resemble the group “disgust towards meat and moral internalization”, identified by Graca et al. [38,57]. Correspondingly, recent evidence suggested a link between low consumption of meat and disgust towards meat based on self-reports and Implicit Association Tests [84]. In line with previous research, both the plant-based eaters and meat-reducers were comparatively high in Openness [41]. Regarding evidence on meat reduction conceptualized as a health behavior, both groups were fittingly higher in Conscientiousness than the medium- to strong-hindrance groups (e.g., [45,47]). Moreover, consistent with previous studies, meat-reducers’ high Conscientiousness was paired with low orientation towards convenience [85]. Despite their high Extraversion, trait expression did not translate into higher reporting of orientation towards sociability. Additionally, differences in Neuroticism were not congruent with the limited body of research [42,43], in that meat-reducers were lowest, but plant-based eaters and medium-hindrance meat eaters showed higher expression than strong-hindrance groups. Mean distributions between groups for both Agreeableness and the Dark Triad supported findings on links between dark traits and health behavior [49].

More to the point, compared to Lacroix and Gifford [10], we discovered a third meat-attached group, the medium-strong-hindrance meat eaters, as a degree between medium- and strong-hindrance meat eaters. While both our medium-strong- and strong-hindrance meat eaters resembled the strong-hindrance group from Lacroix and Gifford [10], ours were characterized by a unique set of inhibitors. Namely, the medium-strong-hindrance group was more attached to meat, whereas the strong-hindrance meat eaters were least proficient in preparing plant-based meals and least environmentally concerned, but most food-neophobic. The inclusion of food consumption orientations provided us with additional insight into motivational differences between groups with low to strong hindrance. Specifically, the strong-hindrance meat eaters were highly concerned with the affordability of food, which was previously highlighted as a barrier of meat reduction to be examined [38,86]. Interestingly, orientation towards affect regulation presented itself as a potential barrier in reducing consumption of animal products. Indeed, differences between groups might at least partially be explained by their differences in Neuroticism [87]. Theoretically consistently, the strong-hindrance meat eaters were high in orientation towards appetite but reported an inconsistently low liking for their foods consumed.

Regarding personality, the strong-hindrance meat eaters strongly differed from all other groups with the lowest expression of Openness, Conscientiousness, and Agreeableness. As discussed above, the general trend (low- to strong-hindrance) is in line with prior research for those traits, aside from Openness the medium-strong-hindrance meat eaters were closer to the low-hindrance groups than the strong-hindrance meat eaters. These effects were even larger for the Dark Triad. Intriguingly, the strong-hindrance meat eaters reported lower consumption of animal products, especially meat, and higher consumption of plant-based foods and alternatives than the medium-strong-hindrance group. Considering evidence on social desirability and self-monitoring [88], it is likely that those strongly hindered meat eaters concealed their meat consumption as part of an interplay between dark traits. Congruously, the strong-hindrance meat eaters showed inconsistencies in their reported willingness to reduce consumption of animal products. While they were comparatively unwilling to reduce meat, dairy and egg consumption, their willingness to adopt a solely plant-based diet was inconsistently high. This explanation is also in line with their high expression of food neophobia. Another reason for their lower reporting of meat consumption might be their genuine efforts to reduce meat and increase meat substitute consumption driven by narcissism, since pro-environmental behavior was found to be associated with (communal) narcissism [89].

With our findings, we fuel the launching debate on how personality, particularly the Dark Triad, constitutes a barrier in reducing meat consumption. Namely, low Agreeableness, Conscientiousness, and Openness and high dark trait expression were identified as inhibitors in a small, strongly hindered subset of our sample. This personality profile was accompanied by low self-efficacy and environmentalism, high price and weight consciousness, as well as a food orientation towards tradition and affect regulation.

Regarding our study design, the share of vegetarians and vegans was higher in our sample than in recent reports on meat consumption in similarly aged Germans and German students, even after accounting for female predominance [90,91]. Missing quota sampling due to sample size concerns and plant-based eaters’ potential bias to participate may have resulted in their greater share. In turn, this might at least partially explain the large profile of plant-based eaters and small profile of strong-hindrance meat eaters estimated by LPA. Future research should investigate whether these profiles are stable across non-student, male-dominant samples with designs accounting for homogeneity [92] as well as a potential distortion of meat consumption due to self-monitoring and social desirability in dark personalities (e.g., deception within fictional scenarios). Furthermore, it should be noted that we did neither exclude participants with medical or religious dietary restrictions nor assess them. Hence, it remains unclear whether individuals with dietary restrictions are characterized by unique sets of hindrances. Prospectively, contextual factors regarding sociability (e.g., dietary habits and attitudes of cohabitors) should also be considered in identifying segments. In general, more simulation studies and derived guidelines for researchers are needed regarding a priori power analysis in order to perform mixture modeling on a large number of indicators without computational restrictions for more complex models.

### 4.2. Implications for Interventions

Expanding results from previous segmentation studies on hindrances of meat reduction, we identified five dietary groups and revealed differences in both inhibitors and facilitators as well as personality. For instance, groups differed in their reported environmental attitudes, self-efficacy, food consumption towards affect regulation and social norms, and dark trait expression.

These group-specific dietary motivations and personality profiles can inform targeted meat-reduction interventions. Factoring in descriptive results, members of all groups were concerned with healthiness and convenience of food preparation. Hence, interventions may provide both plant-based recipes easy to prepare and information on plant-based-friendly (take-out) restaurants. The latter may not be applied to strong-hindrance meat eaters due to their high concerns for price. Regarding food choices, the inclusion of meat replacement recipes might be of particular use in strong-hindrance meat eaters, because of their (reported) efforts to reduce meat via plant-based alternatives and high food neophobia. Moreover, recipes substituting animal protein with legumes, should be considered. Aside from recipes, lack of capability (i.e., self-efficacy) in medium- to strong-hindrance meat eaters may be tackled with other practical resources (e.g., [online] cooking courses) as well as national food guidelines to increase nutritional knowledgability (e.g., [93]). In medium- and strong-hindrance meat eaters, intervention techniques targeting dietary affect regulation might also offer a point of vantage.

Regarding personality, using persuasive appeals [18] may be prolific in targeting strong-hindrance meat eaters, in that arguments for meat reduction rephrased to match needs of communal narcissism could spark reflective motivation [29]. Additionally, volitional personality change may also be applicable in this context [94]. Based on our results, meat reduction may be conceptualized as trait- or more so profile-specific behavior (of high Agreeableness, Conscientiousness, Openness, and low Dark Triad). Correspondingly, recent evidence suggests that personality interventions targeting Agreeableness also reduce expression in the Dark Triad [95].

Aside from supporting plant-based eaters’ and meat-reducers’ existing behavior intentions in reducing meat [9], targeting the consumption of eggs and dairy provides a notable target for future interventions. While meat’s carbon footprint exceeds that from dairy and eggs [25], reported consumption was higher for dairy and eggs in our sample. It is yet to be investigated, whether targeting hindrances of meat reduction also translates into reduced consumption of dairy and eggs. Matching persuasive appeals to Agreeableness (e.g., anthropomorphisation of farm animals) and Openness (e.g., including a wide variety of recipes) may also be considered in intervention design, particularly to reduce consumption of dairy and eggs in plant-based eaters and meat in meat-reducers.

In students specifically, targeting medium- and medium-strong-hindrance meat eaters may yield the greatest benefits, due to their share in our sample and less extreme expression of inhibitors, yet highest reported consumption of meat. Social comparisons could be used to target their high orientation towards tradition and social norms (e.g., [96]). Providing information on low caloric density of plant-based foods might help to confront concerns of weight control. We expect medium-hindrance meat eaters to be more open to altruistic arguments for meat reduction (e.g., to reduce an individual’s carbon footprint) than strong-hindrance meat eaters, considering their higher pro-environmental attitudes. However, based on the reported willingness to reduce meat consumption and willingness to adopt a solely plant-based diet as well as meat attachment between groups, interventions should focus on incorporating replacements for specific foods instead of aiming for prolonged periods of meat abstinence altogether (e.g., meatless days). For example, individual goal setting could revolve around substituting processed meats on sandwiches with plant-based alternatives or chicken breast with legumes on a prototypical plate.

## 5. Conclusions

In conclusion, we identified five dietary groups based on inhibitors and facilitators of meat reduction and highlighted important differences in the consumption of animal products, barriers, consumption orientations and personality. With that, we expand previous segmentation studies by two segments [10,57], providing a more nuanced description of dietary groups for tailoring in future interventions. The present findings suggest that tailoring interventions to not only previously considered barriers (e.g., meat attachment, food neophobia and self-efficacy; [9]), but also personality is worthwhile to potentially improve the efficacy of future interventions. Such work may fuel our knowledge regarding the targetability of personality in the context of dietary change interventions, in which high dark trait expression is expected to constitute a barrier based on our initial results using latent profile analysis. In favor of climate change mitigation, interventions may primarily aim for meat reduction in medium- to strong-hindrance meat eaters and shift focus towards the reduction of dairy and eggs in more plant-based eaters.

## Figures and Tables

**Figure 1 ijerph-19-09354-f001:**
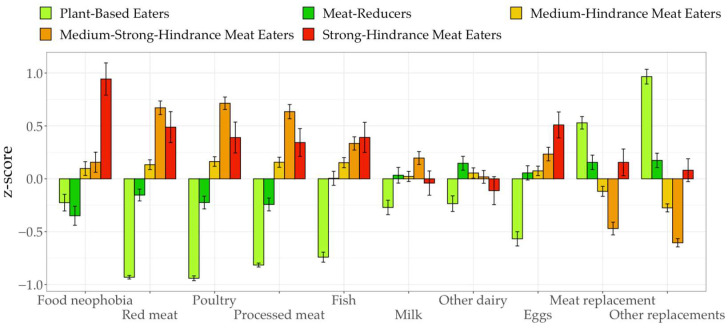
Means of outcome variables between profiles.

**Figure 2 ijerph-19-09354-f002:**
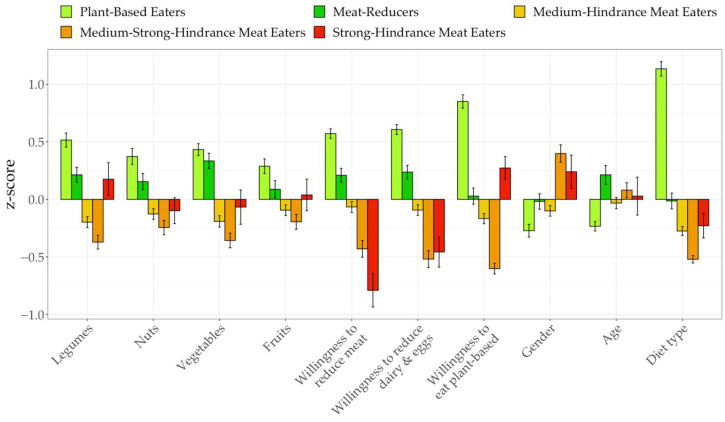
Means of outcome variables between profiles (continued).

**Figure 3 ijerph-19-09354-f003:**
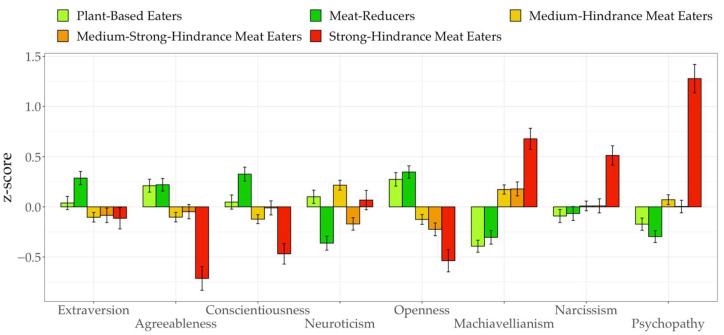
Mean differences in personality between profiles.

**Table 2 ijerph-19-09354-t002:** Model fit indices of latent profile analysis solutions (VVE).

Number of Profiles	*BIC*	*BLRT*	*Entropy*
1	−40,419		
2	−40,214	p<0.01	0.99
3	−40,032	p<0.01	0.90
4	−40,094	p<0.01	0.84
5	−39,968	p<0.01	0.86
6	−40,016	p<0.01	0.85

*Note.* BIC: Bayesian information criterion; BLRT: Bootstrap likelihood ratio test.

## Data Availability

All presented data are publicly available in an online repository (https://osf.io/xewh8/ [accessed on 13 July 2022]).

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
