# Peer review of "You Are What You Eat and So Is Our Planet: Identifying Dietary Groups Based on Personality and Environmentalism"

_ijerph, 2022, doi:10.3390/ijerph19159354_

Round 1

Reviewer 1 Report

This paper reports a study designed to (1) identify dietary groups characterized in terms of their inhibition to reduce meat consumption, and (2) describe the personality traits of those groups.  Participants for the study (n=1135) were recruited online via social media (specific platforms not specified) from a university and a medical school in Germany.  The study identified 5 dietary groups “plant-based eaters”, “meat reducers”, “medium-hindrance meat eaters”, “medium-strong-hindrance meat eaters” using standardized survey tools and characterized personality of these groups.  Lastly, the authors consider the implications of their findings for targeting interventions. It is an interesting study.

General concerns

1.       A stated aim of the authors was to base the study on a “representable” (representative?) sample (line 181). However, representativeness is questionable. It is a student sample and largely female (75%). Most of the participants self-described as were omnivores (64%) and the remainder as vegetarian (29%) or vegan (8%) and it is not clear if that distribution of dietary patterns is representative of German students in general. This should be clarified if the data are available. Further, the  messaging used in recruitment was not mentioned and could have biased the sample in one or more ways.

2.       A general problem throughout is a lack of definitions of terms. For example, in reference to psychological traits they refer to Big Five and the dark traits (line 51). These are defined in the methods section (pages 5 and 6) but it would help the general reader to understand their meaning at first use. They are specialized terms, jargon.  Acronyms should be defined at first use. There are a number of those.

3.       Methods, line 213. Here and elsewhere the authors use to future tense “will” to describe the methods used. The method section should be written in the past tense because it describes what was done, not what was going to be done.

Specific comments

1.       Figures 1 and 2. The acronyms used here (PBE, MR, MHM, MSHM, SHM) should be defined in the figure caption. The reader can guess what they are from the text, but the figure should be intelligible without reference to the text.

2.       Figure 2. I had trouble understanding this figure. The  definitions in the “note” should be part of the figure caption. I was not sure what the “note” was about when I first looked at the figure.

3.       Line 298. The word “both” is ambiguous because the sentence and sentences following focus on only one type of analysis, i.e. LPA.

4.       Table 1. Unclear which of the bulleted points are linked to which of the references cited.

5.       Paragraph that begins in line 57 seems unnecessary.

Line 47 . Use “to” meat reduction rather than “of” meat reduction.

Author Response

Dear Reviewer,

Thank you for your positive overall evaluation. Your inputs improved our manuscript and we responded to all suggestions and edited the manuscript accordingly. “Track changes” was used within the manuscript and responses are summarized in the response letter attached.

Reviewer 2 Report

The scientific article "You are what you eat and so is our planet: Identifying dietary 2 groups based on personality and environmentalism" is of particular interest to specialists who study the problems of organizing a healthy diet, a healthy lifestyle and environmental safety. A well-organized experiment confirms the stated hypothesis. The results of the scientific article are presented logically and accessible.

Author Response

Dear Reviewer,

Thank you for your overall positive evaluation regarding our manuscript.

Sincerely,

the team of authors

This manuscript is a resubmission of an earlier submission. The following is a list of the peer review reports and author responses from that submission.